# Fish Farmers' Perception of Ecosystem Services and Diversification of Carp Pond Aquaculture: A Case Study from Warmia and Mazury, Poland

Konrad Turkowski 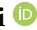

Institute of Management and Quality Sciences, University of Warmia and Mazury in Olsztyn,
ul. M. Oczapowskiego 4/318, 10-720 Olsztyn, Poland; kontur@uwm.edu.pl;
Tel.: +48-8-9523-4332 or +48-6-6335-2912

**Abstract:** Ecosystem services, multifunctionality and diversity play a particular role in the management of carp pond aquaculture. These three concepts have been increasingly considered in sustainable aquaculture science and policymaking. However, the understanding and acceptance of sustainable aquaculture by fish farmers is a prerequisite for the successful implementation of such a targeted policy. Research on a fish farmers' perception of ecosystem services, multifunctionality and diversification of carp aquaculture was carried out in Warmia and Mazury, Poland. The results of interviews showed that farmers have a deep understanding of the ecosystem services and multifunctionality of carp ponds. Production services were indicated as the most important, but the role of ponds in the preservation of biodiversity was another highly valued service. The greatest diversity of the activities, and conviction of their significant impact on the profitability of carp pond farming, was observed on farms with ponds of 1 to 50 ha. In the case of larger farms, the assessment of such impact was more moderate. All surveyed farms provided educational services regarding the ecological values of the ponds. All pond users, regardless of their size, highly rated the need for financial support for the conservation and development of biodiversity.

**Keywords:** carp pond management; ecosystem services; multifunctionality; diversification

## 1. Introduction

Aquaculture, or farming fish and other aquatic organisms, is an important and rapidly growing food-production industry around the world [1]. The EU's Blue Growth Strategy identifies aquaculture as a sector with a high potential which could boost economic growth across the continent and bring social benefits through new jobs [2,3].

Modern aquaculture should be developed as a system to increase product quantity and quality while preserving the environment, considering economic robustness, development of high-quality jobs and building of new relationships among producers, consumers, and the production systems and their associated products [4]. There are two main sections of inland aquaculture in Europe: extensive traditional carp farming in earth ponds and intensive trout farming in different productive facilities. In the case of intensive trout farming, as in the case of majority of intensive forms of aquaculture in the world, the ecosystem approach consists primarily in minimizing the adverse impact on the environment [5,6]. The situation is different in the case of traditional carp pond farming, which has been developing in Central European countries for centuries [7–9].

Traditional carp production is based on the use of natural food with supplementary feeding with cereals. The contribution of supplementary feed to the carp growth gain amounts to around 25–50% of the total yield [10]. This semi-extensive fish production averages around 450–500 kg of fish per ha [11]. Owing to the open nature of the ponds, their integration with the local water body system and the natural feed-based management, such ponds have become integral parts of the local environment. Ponds play important

landscape-related functions, especially in areas with no natural surface waters. Apart from fish production, they supply many services, such as protection against floods, increasing water retention, creating a microclimate and increasing biodiversity [12–14]. A considerable number of ponds are situated in Nature 2000 sites [15–17] and in those classified as nature reserves and protected landscape areas [9,13,14], whose specific features developed under the influence of active fishponds. All these benefits enable the ponds to be defined as an aqua-ecosystem, i.e., a human-managed aquatic ecosystem oriented towards the provision of ecosystem services [4].

The origins of the ecosystem services are to be found in the late 1970s, as a joint initiative of economists and ecologists [18]. They stressed that the valuation of nature services in economic decisions could correct an incorrect assessment of the relationship between man and nature. The most universal definition of ecosystem services was proposed by Costanza et al. [19] as: "the benefits that mankind gains directly or indirectly from the ecosystem functions". Much more complex systems of the service classification have been developed since then, including the UN Millennium Ecosystem Assessment [20] and the Common International Classification of Ecosystem Services—CICES [21].

Fishponds provide ecosystem services similar to those generated by natural wetlands and shallow lakes [14,22,23]. However, unlike natural water bodies, water and trophic conditions in the ponds are controlled and modified by fish farmers. Therefore, the above-mentioned services more closely match the definition of non-commodity outputs in the concept of multifunctional aquaculture [24,25] than services using the concept of ecosystem services regarding natural ecosystems [20].

Agriculture as well as aquaculture is considered multifunctional when it can produce various non-commodity outputs or has other additional functions for food production [26]. Multifunctional agriculture and ecosystem services emerged during the same period and both were recognized as two important concepts for sustainable agricultural development and agricultural policymaking. The two concepts are closely related through their use of the term "function". However, multifunctional agriculture considers functions as agricultural activity outputs and prefers farm-oriented approaches, whereas ecosystem services consider ecosystem functions in the provision of services and prefer service-oriented approaches [27]. In other words, in multifunctional fish farming, all the commodity and non-commodity functions of the fishpond are the result of deliberate management. According to Békefi and Váradi [24], a better understanding of the principle of multifunctional aquaculture and the more systematic application of the various elements of multifunctionality by farmers may contribute to the better placement of pond fish farms in an agroecosystem and improve their viability in the long term. Multifunctionality is often equated with diversification, defined as the adoption or wider application of new species, breeding types and cultural systems [28], as well as various non-fish farming activities, such as the operation of angling ponds, small shops, restaurants or even hotels (depending on the business opportunities) [24,25,29,30]. However, multifunctionality is a broader concept than diversification. In addition to activities/functions that are important from the point of income creation, it also includes all external effects and non-commodity outputs of actions taken, including negative ones. In simplified terms, the multifunctionality of fishponds can be defined as ecosystem services viewed from the perspective of aquaculture farms. However, the term "ecosystem services" has been widely used in aquaculture literature [4,8,31,32] and in EU legal regulations [33], and therefore it is also used in this article.

The implementation of the conservation objectives is associated with several limitations in the management of ponds, lower fish production, higher fish losses and even damage to ponds caused by protected animals. The European Maritime and Fisheries Fund (EMFF) supports aquaculture methods in line with specific environmental needs and subject to specific management requirements resulting from the designation of Natura 2000 sites, as well as aquaculture activities involving the protection and improvement of the environment and the traditional characteristics of aquaculture. Support is provided in

the form of annual compensation for the added costs incurred and/or income foregone [33]. So far, it is the only form of payment for ecosystem services in aquaculture.

Based on the literature review, a conceptual framework of the theoretical relationship between ecosystem services, multifunctionality and diversification of the traditional carp pond aquaculture was developed (Figure 1).

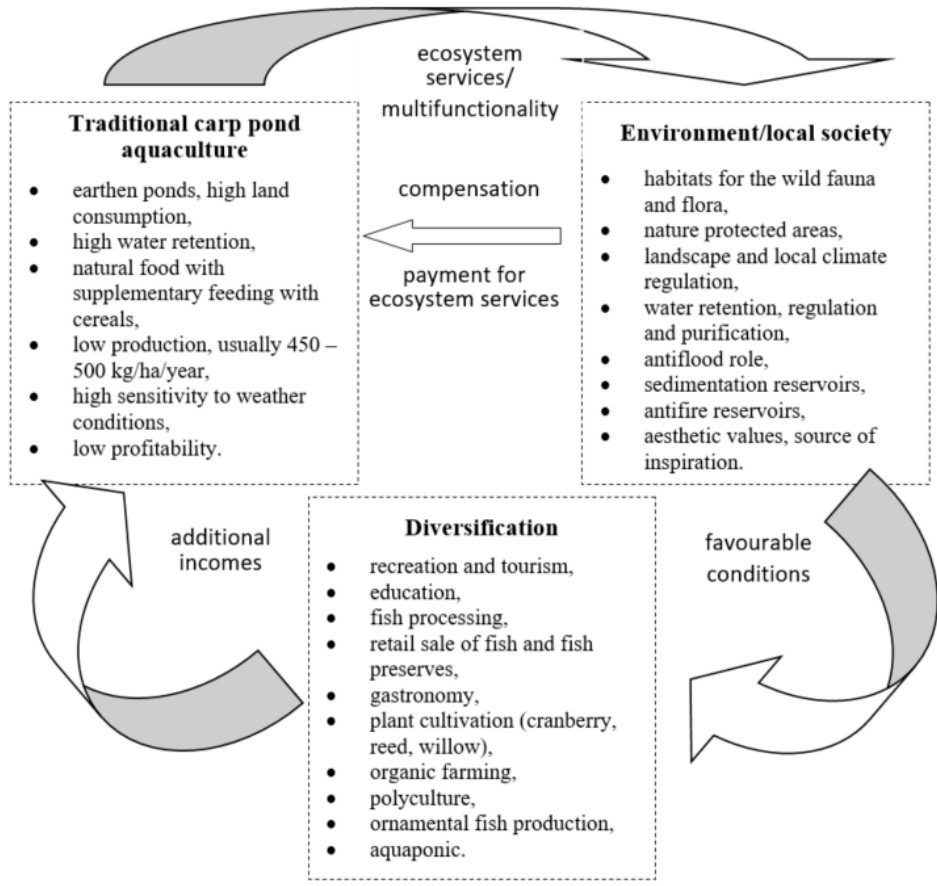

**Figure 1.** Conceptual Framework. Own compilation based on literature review.

Several articles on ecosystem services in aquaculture have recently been published [4,31,32]. However, very few incorporated fish farmers' perceptions of pond ecosystem services [8,25]. These studies allow a better understanding of the motives of fish breeders' decisions and actions and understanding the factors which affect the acceptance of required limitations in fish management. There is another important reason for such research—developing multifunctional sustainable aquaculture requires the understanding and acceptance of those involved in aquaculture.

Starting from the theoretical framework (Figure 1), fish farmers' knowledge, perception and expectation of multifunctional pond management were studied. Semi-structured in-depth qualitative interviews were conducted to answer the following questions: How is the current state of carp pond aquaculture? What opportunities are there for the diversification of commercial activities? Which ecosystem services are important from a farmer's perspective? Is the European instrument (compensating costs and/or lost income) accepted and seen as effective in making management decisions in aquaculture providing environmental services?

## 2. Materials and Methods

### 2.1. Study Area

The area of carp ponds in Poland is the largest in the EU and is estimated at approximately 70,000 ha [34]. This does not include small rural ponds in size usually less than

1 ha, whose total area is estimated in the country at 12,000–15,000 ha. Annual domestic production of carp range between 15,000 and 20,000 mt [34,35]. Poland is the main European market for fresh carp, with a stable consumption of more than 21,000 mt [9]. The carp aquaculture preserves its traditional character with low intensification of production (up to 1500 kg/ha), and a large share of natural food (growing in the pond) in fish diet [30]. Rearing common carp usually employs mixed species stock (polycultures). The contribution of other cyprinids in the final production of carp amounted to 10%, and other freshwater species accounted for an additional 2% [34]. The so-called Dubisch system is commonly used in carp breeding. It facilitates fish farming of the high quality and desired genetic features [30].

The studied region of Warmia and Mazury is in north-eastern Poland (Figure 2).

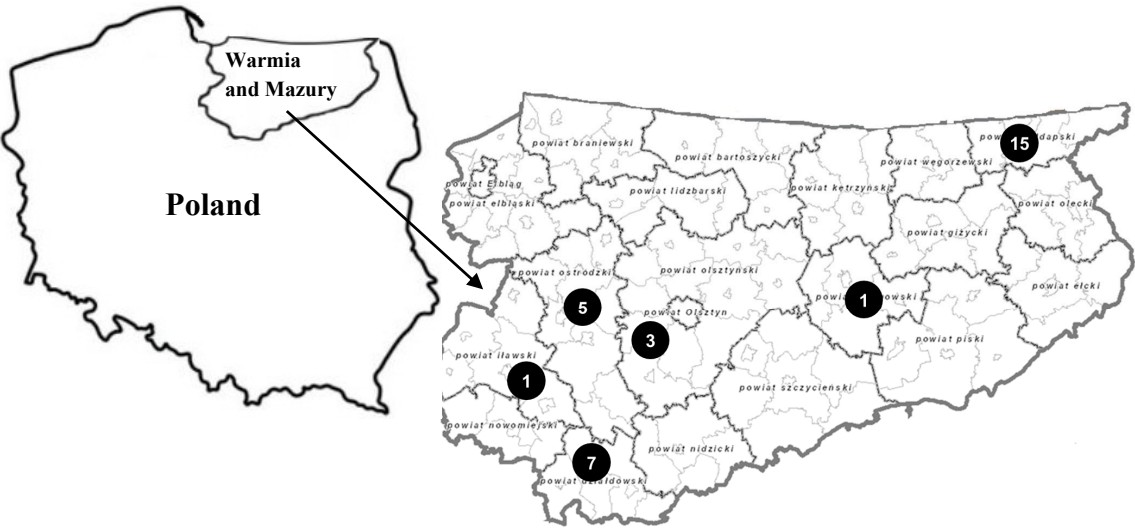

**Figure 2.** Ponds' locations and the number of respondents (n = 32). The counties with the largest number of ponds and their area in the region were selected for the research.

The main landscapes of this study area are croplands (54.4%) and forests (32%). The region also has the largest area, 115,361 ha, of inland waters (4.8%) in Poland, mainly lakes, and a low population density ratio of 1 km$^2$ (59 persons, on average 123 in Poland). There are 16 special protection areas in the region along with 44 special sites for the conservation of habitats belonging to the Natura 2000 network. The area of commercial fishponds in the region totals approximately 1850 ha. Other small ponds and similar water bodies used mainly in agricultural farms for amateur fish breeding are estimated to total another approximately 1000 ha.

### 2.2. Data Collection and Analysis

Thirty-two semi-structured expert interviews [36,37] were conducted with 32 farmers of the Region of Warmia and Mazury (Figure 2). They represent both small family agriculture farming and large fisheries enterprises dealing with fishing in ponds and lakes.

Initial data analysis showed that size had a significant impact on organization and management of the ponds. Due to the total area of the ponds, four classes of farms were distinguished: (a) farms with ponds up to 1 ha, (b) from 1 to 10 ha, (c) from 10 to 50 ha and (d) from 50 ha and more (Table 1).

The author conducted all interviews personally. The interviews began with a short description explaining the goals and background of the study. The average time of the interviews was about 45 min. The shortest interview took about 30 min, the longest about 2 h, including a tour of the ponds and their surroundings. Although representatives of large aquaculture farms were called by phone and invited to participate in an interview, most of the smaller ponds were identified in the field. All farm owners and workers

responsible for pond management were asked to participate in the research. Therefore, the number of respondents exceeded the number of farms (Table 1).

**Table 1.** Selected characteristics of farms and respondents by pond size class.

| Size Class of Farm Ponds | Up to 1 ha | From 1 to 10 ha | From 10 to 50 ha | From 50 ha and More |
|---|---|---|---|---|
| Number of farms | 7 | 6 | 7 | 4 |
| Number of respondents | 7 | 8 | 10 | 7 |
| Total registered area of ponds (ha) | 3.10 | 29.40 | 180.70 | 542.00 |
| Average farms' ponds area (ha) | 0.44 | 4.90 | 25.81 | 135.50 |
| Share of respondents with secondary or higher education in fisheries | 0% | 28% | 33% | 100% |
| Organizational form of the farm: | | | | |
| individual (family) agriculture farm | 7 | 4 | 3 | - |
| individual (family) aquaculture farm | - | 2 | 3 | - |
| limited liability aquaculture and lake fisheries company | - | - | 1 | 4 |
| Average yield (kg of fish per 1 ha) | 258 | 550 | 1010 | 729 |
| Main markets and forms of sale of fish | for own needs (not for sale) | fishing grounds for anglers, local, domestic, and foreign market | | |

The interview concerned the types of commercial activities carried out on the ponds, their diversity and the possibility of integrating carp pond management with other sectors, as well as an assessment of the impact of European fisheries' strategy measures. When answering the questions about diversification of the activity carried out on the ponds and the possibility of its integration with other forms of activity, a respondent could choose the options applicable to his/her farm from those proposed. A respondent could also indicate a different form, not mentioned in the questionnaire (Appendix A).

The part of the interview was based on closed questions and included questions that revealed preferences in the ranking of selected potential pond ecosystem services.

The ecosystem services survey contained a list of 29 potential ecosystem services presented in alphabetical order, including 8 production, 12 environmental and 9 social services. They corresponded to the classification of pond ecosystem services used by Mathé and Rey-Valette [8] and Popp et al. [25]. The respondents rated the importance of each service, assigning it points from 0 (not important) to 4 (very important). The resulting rankings were presented in percentage terms.

### 3. Results and Discussion

*3.1. What Kind of Commercial Activity Is Carried Out on the Ponds?*

Production of fish for consumption and of fish stocking material was conducted in all the farms' ponds. Ponds with a total area of more than 1 ha were used by farmers for commercial purposes. The greatest diversity of the activities was observed on farms with ponds of 1 to 50 ha, which—apart of fish production and fish processing plants—produced ornamental fish and had special fishing sites for anglers. The main activity of farms with ponds over 50 ha was mainly limited to fish and stocking material production and fish processing (Figure 3).

In all farms, additional activities were carried out that were not directly related to fish production, but were conditioned by ecological and other values, contributing to the ecosystem services of the ponds (Figure 1). Educational services of the widest range were offered on farms with ponds of 10 to 50 ha. There were educational paths and birdwatching sites on 50% of the farms and overnight school trips were organized in 17% of them. Fish



frying points, as well as agri- and eco-tourism activities, were conducted on farms with ponds over 10 ha (Table 2).

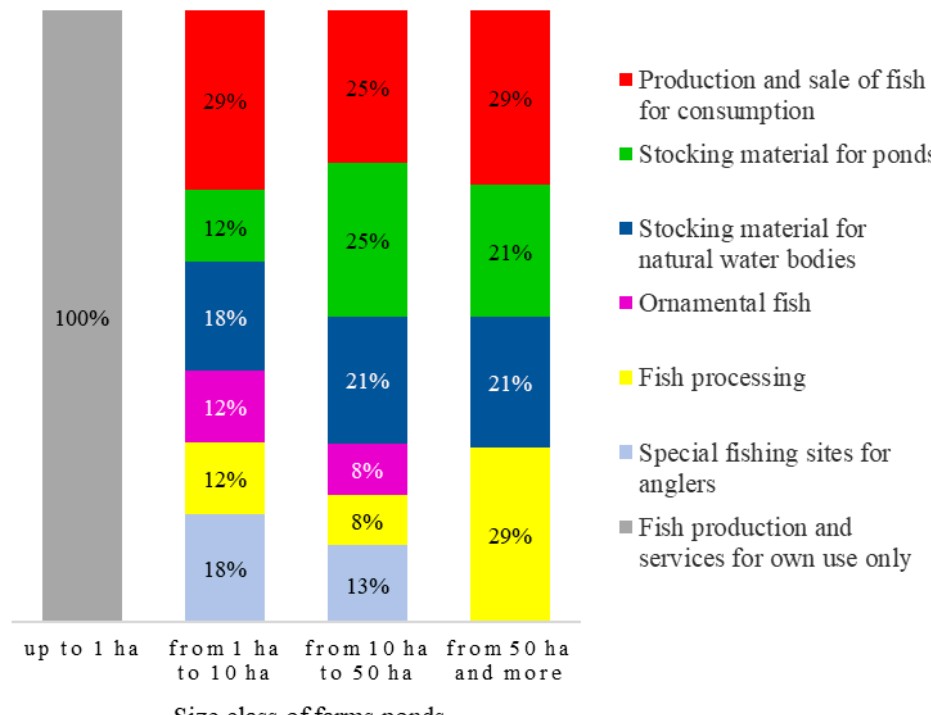

**Figure 3.** Structure of the main aquaculture activity on farms with different pond size classes (n = 24).

**Table 2.** Additional activities carried out in the studied farms with ponds over 1 ha (% of the number of farms, n = 17).

| Specification | From 1 to 10 ha | | From 10 to 50 ha | | From 50 ha and More | |
|---|---|---|---|---|---|---|
| | Implemented | Planned | Implemented | Planned | Implemented | Planned |
| Schools' trips | 14% | - | 17% | 17% | - | - |
| Educational paths | 14% | 29% | 50% | 17% | 25% | - |
| Birdwatching sites | - | 14% | 50% | 17% | 25% | - |
| Fish frying point | - | 14% | 33% | 17% | 25% | - |
| Agri- and eco-tourism | - | - | 33% | 33% | 25% | - |
| Water sport equipment rental | - | - | - | 17% | - | - |
| Conference Organization | - | - | - | 33% | - | - |
| Fishing museum | - | - | - | 33% | - | - |

Farms with ponds of 10 to 50 ha conducted the largest number of additional activities and planned to develop such activities to the greatest extent, including water sport equipment rental, holding fishing conferences and museums, i.e., Table 2. The relatively small attention (absence in the farm owners' plans) devoted by farms with ponds of over 50 ha to additional activities can be attributed to the focus being placed on main commercial fisheries activity. The little interest in the diversification of activity can also be explained by the structure of these farms. As pond and lake fisheries companies (Table 1), they also managed several thousand hectares of lakes, which sufficiently diversified their combined fisheries activities.

The respondents pointed to a moderate possibility of integrating the use of carp ponds in managing waste from intensive fish fattening (carp, African catfish, etc.), which would

be used as fertilizer in carp ponds (Table 3). A total of 24% of respondents from all pond farms indicated this possibility.

**Table 3.** Possibility of integrated use of carp ponds with other forms of farming (% positive answers, n = 25, including farms' representatives with ponds from 1 to 10 ha, n = 8; from 10 to 50 ha, n = 10; from 50 ha and more, n = 7).

| Specification | From 1 to 10 ha | From 10 to 50 ha | From 50 ha and More | Total |
|---|---|---|---|---|
| Irrigation of agricultural crops | 13% | 20% | - | 12% |
| Managing waste from intensive fish fattening (carp, African catfish, etc.) | 13% | 50% | - | 24% |
| Aquaponics (greenhouse cultivation of plants using water from fishponds) | 13% | 30% | - | 16% |
| Cultivation of cranberries | 13% | 30% | - | 16% |
| Growing energy crops | - | - | - | - |

The largest number of such responses were given by respondents from farms with ponds of 10 to 50 ha, which concerned using wastewater (50%), aquaponics (greenhouse cultivation of plants using water from intensive fish fattening), cultivation of cranberries (30% each) and the use of water for the irrigation of agricultural crops (20%). The same scope of integration was indicated by respondents from farms with smaller ponds of 1 to 10 ha, but only 13% of responses were positive. Respondents from farms with ponds of 50 ha and more did not indicate any possible form of integration of carp pond use with other forms of farming. None of the respondents in the entire surveyed group of farms mentioned the possibility of integrating the fishpond with growing energy crops (Table 3).

### 3.2. How Does Additional Activity Affect the Financial Condition of Pond Farms?

Opinions on the impact of additional activities on the financial situation of pond farms in the total group of respondents under study was divided equally between those who thought that the impact was negligible (36%) and those who thought it had a considerable impact (36%). The others (28%) felt that such activities had a small impact on the farm's finances (Figure 4).

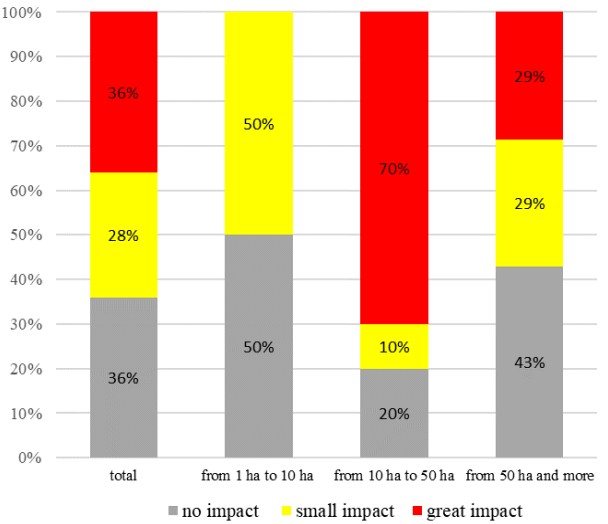

**Figure 4.** The rating of the impact of additional activities conducted at fishponds on the financial situation of the farm as seen by respondents for farms with ponds of 1 to 10 ha (n = 8), from 10 to 50 ha (n = 10), and from 50 ha and more (n = 7).

The rating of the impact of the additional activities on the financial situation largely reflected the scale of such activities of the identified groups of farms. The conviction of a great impact of additional activities (70%) was expressed by respondents who represented

farms with ponds of 10 to 50 ha, with the greatest scale of such activities—both conducted and planned (Table 2). They also highly appreciated the possibility of integrating the carp pond farming with other forms of aquaculture and agriculture activity (Table 3).

### 3.3. Farmers Evaluation of Carp Ponds Ecosystem Services/Multifunctionality

Production of fish and other aquatic organisms were seen as the most important pond services, regardless of their size. The total rating of the services was close to the maximum (Figure 5).

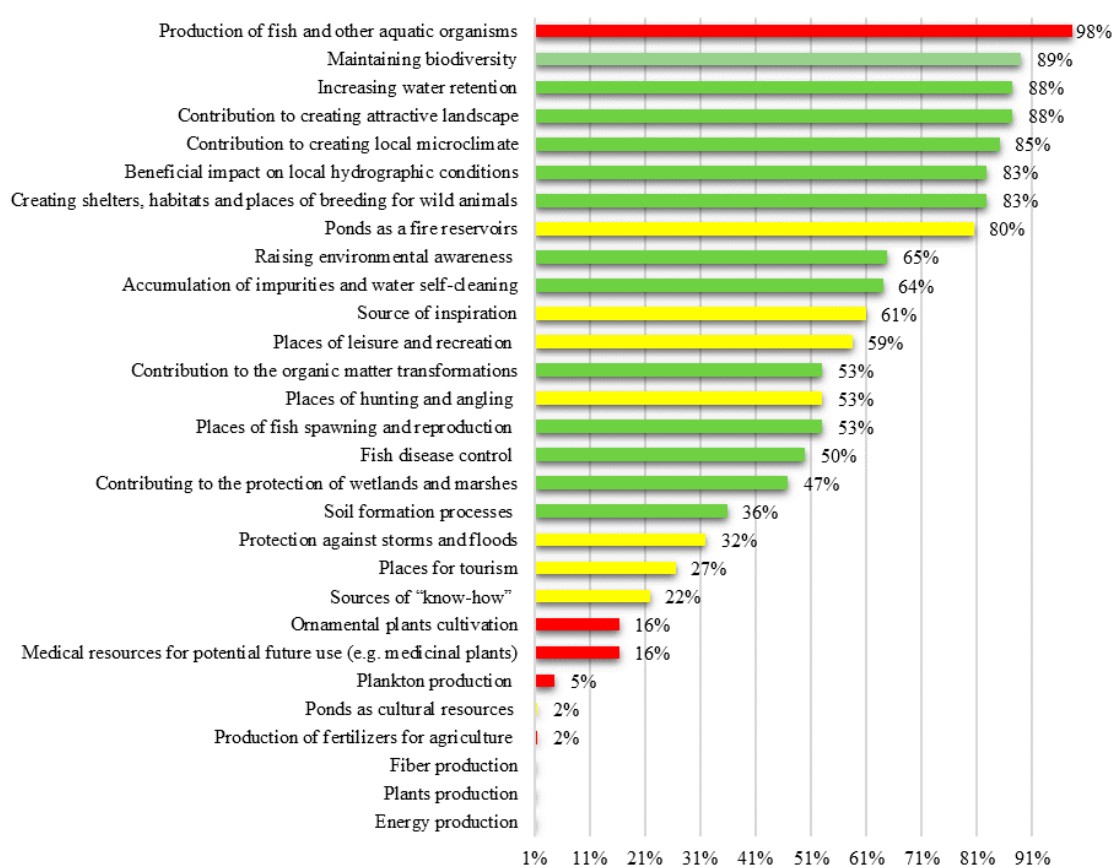

**Figure 5.** Rating of ecosystem services (production—red, environmental—green, social—yellow) provided by ponds in all surveyed farms (n = 32 respondents).

The other production services: medical resources for potential future use (e.g., medicinal plants), ornamental plants, plankton production and production of fertilizers for agriculture were generally thought to be of little importance, whereas production of energy, fiber and plants were regarded as completely irrelevant (Figure 5).

The ratings of the services were diverse in individual pond size groups. While the biodiversity, water retention, contribution to the organic matter transformation and protection of wetlands and marshes were assessed similarly in all studied pond size classes, the other ecosystem services such as contribution to creating places for fish spawning and reproduction, attractive landscape, local microclimate, shelters, habitats and places for wild animals' breeding, as well as accumulation of impurities and water self-cleaning, were assessed significantly lower in farms with the largest area of ponds (Figure 6).

The importance of ponds as fish breeding sites decreased along with the increase in the ponds' size and the importance of stocking material from artificial reproduction, which dominates on large ponds. Owners of the smallest ponds (less than 1 ha) demonstrated a pro-ecological attitude, even though they were not covered by financial support from EU fish programs.

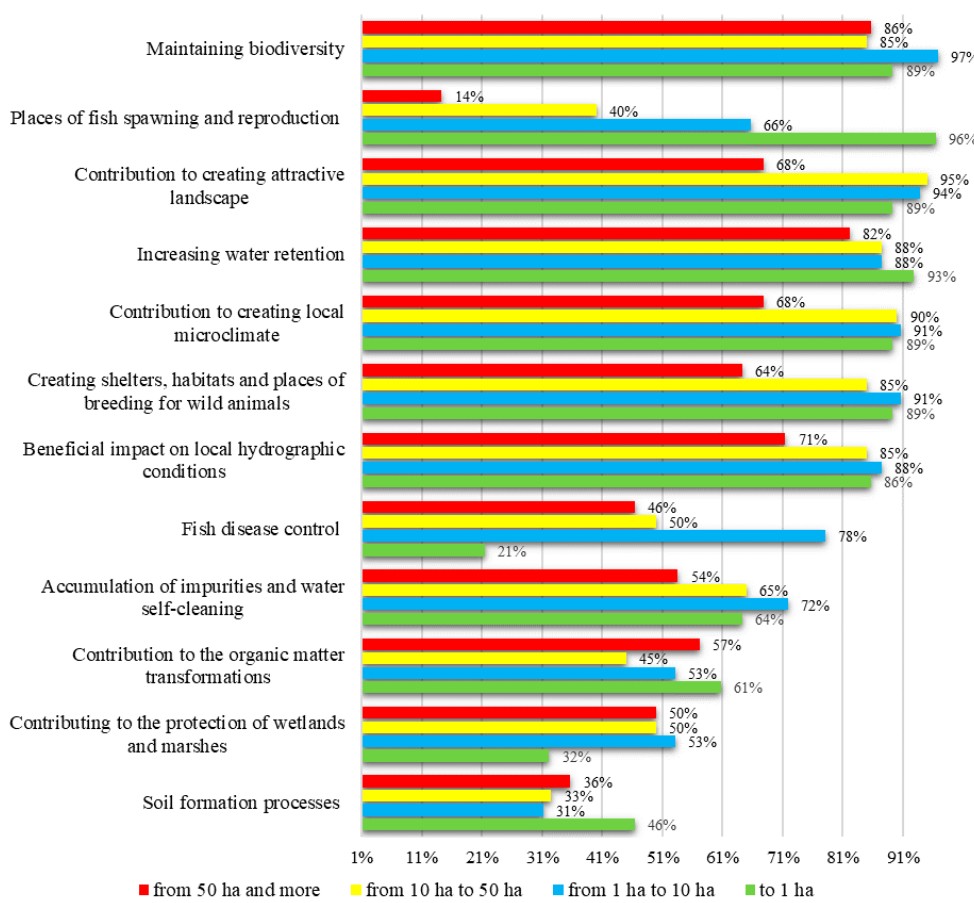

**Figure 6.** Rating of ponds' environmental services in farms with different ponds size classes (n = 32) respondents, including under 1 ha (n = 7), from 1 to 10 ha (n = 8), from 10 to 50 ha (n = 10) and from 50 ha and more (n = 7).

Among the social services (Figure 7), the pond role as a fire tank got the highest rate of 80%. Other social services of ponds as sources of raising environmental awareness (65%) and inspiration (61%), places of leisure and recreation (59%) as well as hunting and angling (53%) received slightly lower ratings. The role of ponds in protection against storms and floods (32%), as tourist destinations (27%) and as a source of "know-how" (22%) was assessed significantly lower. Probably the lack of historical devices was the main reason for the poor assessment of ponds as cultural resources. Social services of smaller ponds up to 10 ha were rated by their owners more than twice as high as the largest group of ponds (50 ha and more). The ratings from fish farmers of ponds of 10 to 50 ha in size were properly balanced.

Farmers with large total ponds area valued them lower as places of recreation, hunting and fishing, as well as a source of "know-how", environmental awareness and inspiration, while these services received high marks from farmers with smaller ponds. The exception was protection against storms and floods, which can probably be attributed to the better ability to provide these services by larger ponds. The highest rating of the ponds as source of inspiration (83%), place of recreation and tourism (55%), as well as sources of "know-how" (38%) for farms with ponds of 10 to 50 ha (Figure 7) is reflected in expanded additional activities (Table 2) and in the opinion of their significant impacts on the farms' income (Figure 4).

A similar general order of importance of ecosystem services: productive, environmental and then social, has been found in French [8] and Hungarian [25] studies. Production services were indicated as the most important, but the role of ponds in the preservation of biodiversity was another highly valued service. The environmental functions were even more preferred by users of Polish ponds. This result can be attributed to the fact that while

the French and Hungarian research concerned large pond farms, the presented study also covered (approximately 30%) agriculture farms with small ponds of less than 1 ha, used exclusively for non-commercial purposes (Table 1).

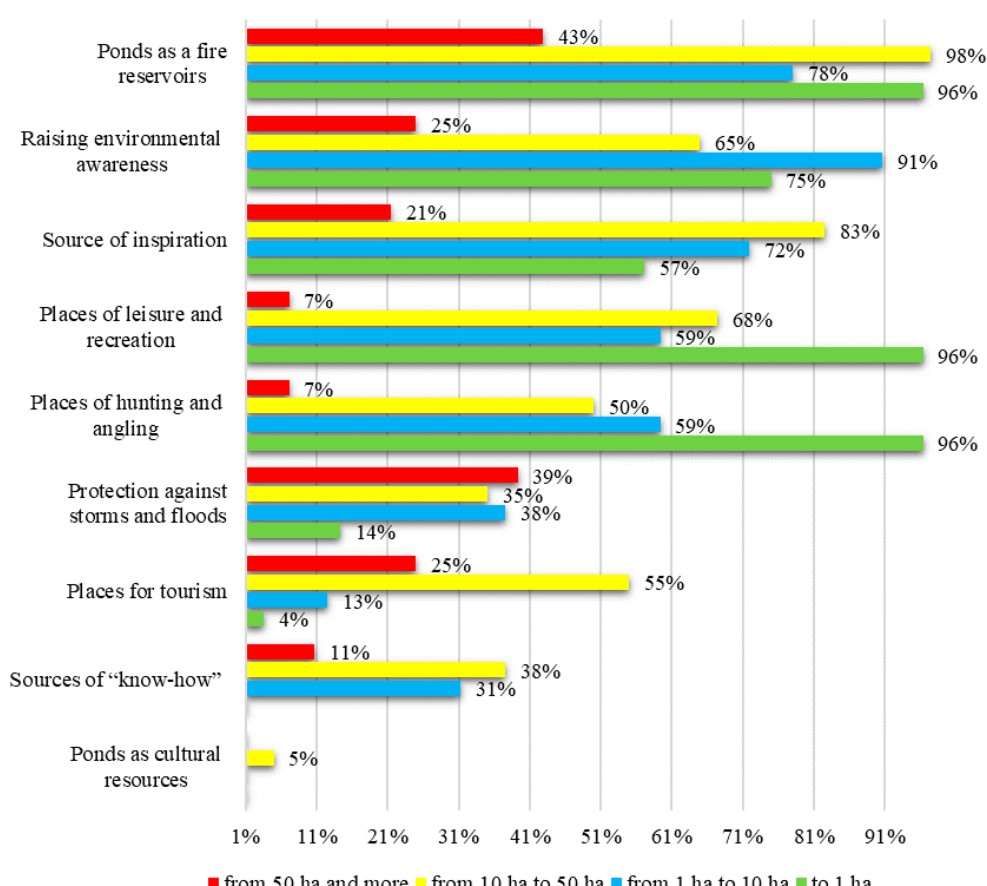

**Figure 7.** Ratings of social services of fishponds in farms with different ponds size classes (n = 32 respondents, including under 1 ha (n = 7), from 1 to 10 ha (n = 8), from 10 to 50 ha (n = 10) and 50 ha and more (n = 7)).

*3.4. European Compensation Instrument for the Costs and/or Income Foregone in Aquaculture Providing Environmental Services*

EMFF supports the development of aquaculture environmental services, especially in terms of Art. 54.1. Aquaculture providing environmental services [33]:

- Aquaculture methods compatible with specific environmental needs and subject to specific management requirements resulting from the designation of Natura 2000 areas.
- Participation, in terms of directly related costs, in ex-situ conservation and reproduction of aquatic animals, within the framework of conservation and biodiversity restoration programs developed by public authorities, or under their supervision.
- Aquaculture operations which include conservation and improvement of the environment and biodiversity and management of the landscape and traditional features of aquaculture zones.

Support for aquaculture in Natura 2000 areas takes the form of annual compensation for the additional costs incurred and/or income foregone as a result of management requirements in the areas concerned. Obtaining support requires meeting certain conditions. Support for the aquaculture operations, which include conservation and improvement of the environment and biodiversity, management of the landscape and traditional features of the aquaculture, is only be granted to beneficiaries who commit themselves for a minimum period of five years to aqua-environmental requirements that go beyond the mere application of EU and national

law. The environmental benefits of the operation shall be demonstrated by a prior assessment conducted by competent bodies designated by the Member State, unless the environmental benefits of that operation have been already recognized.

Users of all studied ponds, regardless of their sizes, highly rated the need to provide financial support for biodiversity preservation and development. Among the group of respondents (22) associated with farms with ponds area of over 1 ha (benefiting from this kind of support), only two, representing farms with ponds of 50 ha and more, were against the system of aqua-environmental measures. None of the respondents rated the current system of aqua-environmental measures as very good. The system was rated as good by 9%, and sufficient by 41% of the respondents, while 27% of respondents considered it to be bad, and 23% as a mediocre system. The low ratings of the current aqua-environmental measures were caused by the over-expanded system of documenting the task performance verification (39% of responses) and its too complicated (33% of responses) and expanded (28%) nature. In general, almost all respondents (80–100%) agreed on the need to introduce compensation for production interruptions that occurred through no fault of fish farmers, caused by fish diseases, drought, water pollution or flooding.

## 4. Conclusions

The study showed that fish farmers are aware that the use of traditional carp breeding methods contributes to the preservation and development of biodiversity and generates other ecosystem services. However, since traditional methods of fish breeding and rearing are not too profitable, the respondents had high expectations regarding expanding and modifying the EU system of aqua-environmental measures. The system of aqua-environmental measures was rated as too complicated and expanded. All fish farmers indicated an urgent need for compensation for production interruptions.

The conducted research showed that the size of the ponds determined the way of their management. The smallest ponds under 1 ha were used to satisfy the needs of the farm owners and their families. Commercial activity was conducted by farms with the total ponds area over 1 ha. It was diversified and included table fish, stocking material and ornamental fish production, then fish processing and providing special fishing sites for anglers. All farms carried out additional activities that were not directly related to fish production but were conditioned by ecological and other values associated with the ecosystem services of the ponds. These activities were undertaken in the field of education, agritourism and gastronomy. The most diverse activities were carried out by farms with ponds ranging from 10 to 50 ha. This size was too small to obtain a satisfactory income from traditional carp production, but it was sufficient to ensure broad opportunities for diversification. Farms with ponds over 50 ha were much less active in this respect. The study revealed a moderate willingness of the farm owners to integrate carp management with other forms of aquaculture or agriculture activity. Again, the farms with total ponds area of 10 to 50 ha were the most open in this respect. They did not exclude integration of carp breeding with the utilization of wastewater from intensive aquaculture, aquaponics or with the cultivation of cranberries and an agriculture system of crop irrigation.

Production of fish was seen as the most important pond function, regardless of their size. The research confirmed a similar general order of importance of carp pond ecosystem services: productive, environmental and then social, which was found in studies of carp pond farms in France [8] and in Hungary [25]. The environmental and social functions of fishponds were generally better rated on farms with smaller ponds. There was a noticeable relationship between the assessment of social ecosystem services and the diversification of activities carried out on the ponds. Ponds as a source of inspiration, know-how and a place of destination for recreation and tourism were rated the highest in farms with 10 to 50 ha of ponds, with the most diversified additional activities.

Research has confirmed that ecosystem services and multifunctionality are recognized and accepted elements of the traditional carp pond aquaculture. The implementation of this

kind of aquaculture is favored both by the European policy and the ecological awareness of fish farmers.

**Funding:** The research was co-financed by the European Union under the national Technical Assistance of the Rural Development Program for 2014–2020 (Poland).

**Institutional Review Board Statement:** Not applicable.

**Informed Consent Statement:** Not applicable.

**Data Availability Statement:** The data presented in this study are available on request from the corresponding author.

**Conflicts of Interest:** The author declares no conflict of interest.

## Appendix A

**Interview questionnaire**

The surveys cover farm managers/owners and users of carp ponds.

Operation: "Unused potentials of sustainable development of rural areas in the Warmia-Masuria Province", co-financed by the European Union under the Action Plan of the National Network of Rural Areas for 2014–2020, Operational Plan for 2018–2019.

**The interviewee**

Job position: . . . . . . . . . . . . . . . . . . . . . . . . . . . . . . . . . . . . . . . . . . . . . . . .

Education: □ primary □ secondary □ secondary in fisheries □ higher □ higher in fisheries

**Farm**

Organizational form of your farm-individual (family) agriculture farm, individual (family) aquaculture farm, limited liability aquaculture and lake fisheries company, other:

. . . . . . . . . . . . . . . . . . . . . . . . . . . . . . . . . . . . . . . . . . . . . . . . . . . . . . . . . . . . . . . . . . . . . . . . .
. . . . . . . . . . . . . . . . . . . . . . . . . . . . . . .

**Table A1.** Carp ponds and fish markets.

| |
| --- |
| Total registered area of ponds (ha) |
| Average annual yield (kg)<br>(for the last three years) |
| Main fish markets (local, regional, domestic) and forms of sale |

**A. Ecological services/multifunctionality**

Please rate each of the ponds' service/function on a scale from 0 to 4, where:

0—means negligible importance in relation to others,

1—moderate importance,

2—comparable importance to others,

3—is clearly more important than others,

4—is definitely more important than the others.

**Table A2.** List of potential carp ponds' ecological services.

| No. | Service/Function of Carp Ponds | 0 | 1 | 2 | 3 | 4 |
| --- | --- | --- | --- | --- | --- | --- |
| 1 | accumulation of impurities and water self-cleaning | | | | | |
| 2 | beneficial impact on local hydrographic conditions | | | | | |
| 3 | contributing to the protection of wetlands and marshes | | | | | |
| 4 | contribution to creating attractive landscape | | | | | |
| 5 | contribution to creating local microclimate | | | | | |

**Table A2.** *Cont*.

| No. | Service/Function of Carp Ponds | 0 | 1 | 2 | 3 | 4 |
|---|---|---|---|---|---|---|
| 6 | contribution to the organic matter transformations | | | | | |
| 7 | creating shelters, habitats, and places of breeding for wild animals | | | | | |
| 8 | energy production | | | | | |
| 9 | fiber production | | | | | |
| 10 | fish disease control | | | | | |
| 11 | increasing water retention | | | | | |
| 12 | maintaining biodiversity | | | | | |
| 13 | medical resources for potential future use (e.g., medicinal plants) | | | | | |
| 14 | ornamental plants cultivation | | | | | |
| 15 | places for tourism | | | | | |
| 16 | places of fish spawning and reproduction | | | | | |
| 17 | places of hunting and angling | | | | | |
| 18 | places of leisure and recreation | | | | | |
| 19 | plankton production | | | | | |
| 20 | plants production | | | | | |
| 21 | pond as a fire reservoir | | | | | |
| 22 | ponds as cultural resources | | | | | |
| 23 | production of fertilizers for agriculture | | | | | |
| 24 | production of fish and other aquatic organisms | | | | | |
| 25 | protection against storms and floods | | | | | |
| 26 | raising environmental awareness | | | | | |
| 27 | soil formation processes | | | | | |
| 28 | source of inspiration | | | | | |
| 29 | sources of "know-how" | | | | | |

**Table A3.** Diversified and integrated carp pond management.

| What Kind of Commercial Activity Is Carried Out on the Ponds? | |
|---|---|
| production and sale of fish for consumption | ☐ |
| stocking material for ponds | ☐ |
| stocking material for natural water bodies | ☐ |
| ornamental fish | ☐ |
| fish processing | ☐ |
| special fishing sites for anglers | ☐ |
| fish production and services for own use only | ☐ |

**Table A3.** *Cont*.

| What Additional Activities Are Carried Out on the Ponds? | |
| --- | --- |
| schools' trips | ☐ |
| educational paths | ☐ |
| birdwatching sites | ☐ |
| fish frying point | ☐ |
| agri- and eco-tourism | ☐ |
| water sport equipment rental | ☐ |
| conference organization | ☐ |
| fishing museum | ☐ |
| others: | |
| **Is It Possible to Integrate Use of Carp Ponds with Other Forms of Farming? If so, Indicate Examples:** | |
| irrigation of agricultural crops | ☐ |
| managing waste from intensive fish fattening (carp, African catfish, etc.) | ☐ |
| aquaponics (greenhouse cultivation of plants using water from fishponds) | ☐ |
| cultivation of cranberries | ☐ |
| growing energy crops | ☐ |
| others: | |
| **What Additional Activities Can Be Undertaken on Your Ponds?** | |
| schools' trips | ☐ |
| educational paths | ☐ |
| birdwatching sites | ☐ |
| fish frying point | ☐ |
| agri- and eco-tourism | ☐ |
| water sport equipment rental | ☐ |
| conference organization | ☐ |
| fishing museum | ☐ |
| others: | |
| **How Do These Additional Activities Affect the Financial Condition of Your Farm?** | |
| great impact | ☐ |
| small impact | ☐ |
| no impact | ☐ |

**Table A4.** European compensation instruments.

| 1. How Do You Evaluate the Current System of Aqua-Environmental Measures? | | | | |
| --- | --- | --- | --- | --- |
| ☐ bad | ☐ mediocre | ☐ sufficient | ☐ good | ☐ very good |
| **2. Please Give Reasons for Negative Assessment of the System?** | | | | |
| is too expanded | | | | ☐ |
| is too complicated | | | | ☐ |
| over-expanded system of documenting the task performance verification | | | | ☐ |
| others: | | | | |

**Table A4.** *Cont.*

| 3. Is It Worth Maintaining the System of Aqua-Environmental Measures? | |
|---|---|
| ☐ yes | ☐ no |

**4. Should Compensation for Interruptions in Fish Production Be Introduced? If so, It Is Because of:**

| | |
|---|---|
| fish diseases | ☐ |
| flooding | ☐ |
| drought | ☐ |
| water pollution | ☐ |
| others: | |
| This kind of compensation should not be introduced. | ☐ |

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
