# Peer review of "Fish Farmers’ Perception of Ecosystem Services and Diversification of Carp Pond Aquaculture: A Case Study from Warmia and Mazury, Poland"

_sustainability, doi:10.3390/su13052797_

Round 1

Reviewer 1 Report

The study deals with the issues of rising importance - sustainability of carp pond farming under changing approaches to ecological background of aquaculture. Undoubtedly, farmers' access to more environmentally friendly and multifunctional management of carp ponds plays a crucial role in this process. For this reason, it is very important to know their opinion about possible future routing of carp pond aquaculture.

Regarding the quality of submitted manuscript, I recommend the following improvements:

  • the English must be improved to make the text better understandable.
  • the crucial problem is in understanding the structure of the farms under study. It must be clearly stated what are we talking about - farms of TOTAL area of their ponds (l. 153) up to 1 ha, 1 - 10 ha, 10 - 50 ha, more than 50 ha? Or does the author mean the ponds of different area? With this respect, some statements are confusing - see e.g. l.378 - "the users of ponds of 10 ha to 50 ha" - does it mean the farms with ponds of 10 to 50 ha or really the ponds of 10 to 50 ha ? Or farms which total area of ponds ranges between 10 and 50 ha? This must be clearly stated in Material and Methods and clearly commented throughout the whole manuscript.
  • How to understand the higher number of respondents than the number of farms interviewed?
  • Minor comments:
  • l. 43 - carp pond culture with the use of supplementary feeding is the way of semi-intensive production, not an extensive one. Extensive aquaculture means no supplementary feeding (and no fertilization)
  • Fig. 2 - figures indicating the number of respondents (green circles) are difficult to read
  • the names of some authors (known to me) in References are wrong, namely in l. 416 Oberle M. and in l.419 - Mössmer M., Hauber M.

Reviewer 2 Report

The work deals with a very relevant theme.
The integration of the concepts of multifunctionality of the investigated
activities with the concept of ecosystem services and diversification
appears really interesting. I suggest to the author to rework the introduction and in particular
to enhance the theoretical framework that is sacrificed in
the introduction but which deserves at least a separate paragraph.

In particular its important to create links between different concepts (line 55-56)
Furthermore, attention must be paid to the part relating to policies
which seems to have been copied from European legislation.
In this case it would be better to put the quotes :
par. 3.4. European compensation instrument for the costs and/or income"

Reviewer 3 Report

Dear Author,

I am writing this to submit my comments on your research article with the following details.

Manuscript title: Ecosystem services and diversification of carp pond aquaculture: a case study from Warmia and Mazury, Poland

Manuscript Number: sustainability-1117390

Journal Submitted: Sustainability

Specific Comments:

Title:

The title is okay.

Abstract:

The abstract has an issue. It should be rewritten by incorporating the results and what are the main conclusions of this study.

Introduction:

The introduction is a bit lengthy. Please consider this.

You also need to discuss a little further about the predominant type of aquaculture currently being run in the carp ponds as well as which are the most important carp species being cultured?

An overview of the Poland carp aquaculture industry is also missing.

Materials and Methods:

Perfectly done. Obviously, no analysis is involved.

However, how about sharing the questionnaire used during the interview? I suggest you sharing the questionnaire so that your paper gets more attention and could be followed easily by researchers in future?

Results and Discussion:

I feel it is well-coordinated and presented nicely.

Figures and Tables:

Figure 1. Text visibility is a problem. Please explain a bit more about the framework in the caption.

Figure 2. The caption calls for more details about the locations.

Conclusions

Conclusions are a bit too detailed. Consider if it could be done more concisely?

References:

Good to go.

Round 2

Reviewer 1 Report

Minor check of English would be beneficial
